

# Crops' sensitivity and adaptive capacity to drought occurrence

Catarina Alonso[1], Célia Gouveia[1,2], Ana Russo[1], Patrícia Páscoa[1]

[1]Instituto Dom Luíz, Faculdade de Ciências da Universidade de Lisboa, Campo Grande
[2]Instituto Português do Mar e da Atmosfera, Lisboa, Portugal

*Correspondence to*: cmgouveia@fc.ul.pt

**Abstract.** In the context of sustainable agricultural management, drought monitoring plays a crucial role assessing the vulnerability of agriculture to drought occurrence. Drought events are very frequent in the Iberian Peninsula (and in Portugal in particular) and an increase of frequency of these extreme events are expected in a very near future. Therefore, the quantitative assessment of the natural ecosystems vulnerability to drought is still very challenging, mainly due to the difficulties of having

a common definition of vulnerability. Consequently, several methods have been proposed to assess agricultural vulnerability. In this work, a principal component analysis (PCA) was performed based on the components which characterize the Exposure, Sensitivity and Adaptative Capacity of the agricultural system to drought events with the aim of generating maps of vulnerability of agriculture to drought in Portugal. Several datasets were used to describe these components, namely drought indicators, vegetation indexes and soil characterization variables. A comparison between the PCA-based method and a

subjective non-automatic categorical method using the same indicators was performed. Results show that both methods identify Minho and Alentejo as regions of low and extreme vulnerability, respectively. The results are very similar between the two methods, with small differences on certain vulnerability class. However, the PCA method has some advantages over the categorical method, namely the ability to identify the sign of the indicators, not having to use the indicator-component subjective relationship, nor calculating weights. Furthermore, the PCA method is fully statistical and presents results according

to a prior knowledge of the region and the data used.

## 1 Introduction

The frequency and severity of extreme weather events have increased notably in recent decades (Field et al., 2012). Studies on climate variability and future projections indicate that weather extremes are and will continue to pose as a threat to agriculture (FAO, 2016), undermining food security and sustainable agriculture (Murthy et al., 2015b). Droughts may produce significant

impacts on agriculture, namely through shortages in water supply, destruction of ecological resources and losses in agricultural production. In more sensitive regions, these impacts may result in hunger, human suffering, death and abandonment of geographical regions ( WMO, 2014; FAO, 2017). Thus, identification, assessment and ranking of drought events is a crucial step towards disaster risk reduction ( Maskrey, 1993;Bogardi and Birkmann, 2005) .

A drought event is commonly defined as a period with abnormally reduced precipitation resulting in water scarcity in the

region (Vicente-Serrano et al., 2010a). However, the scarcity of water is not exclusively due to reduced precipitation. High



temperatures lead to increased soil water demand due to evapotranspiration (Sheffield et al., 2012; Wang et al., 2012), promoting also dryness  (Dai, 2011). Several methods for analyzing drought episodes have been developed and applied, allowing a better monitoring and characterization of drought events. Among the most used approaches, the use of drought indexes such as the Palmer Drought Severity Index (PDSI) (Palmer, 1965), the Standardised Precipitation Index (SPI) (McKee

et al., 1993) and the Standardised Precipitation-Evapotranspiration Index (SPEI) (Vicente-Serrano et al., 2010a, 2010b) are frequently referred, each one presenting advantages and weaknesses in its application. The SPEI combines the thermal sensitivity of the PDSI, i.e. the effect of temperature on evapotranspiration, with the multi-scalar nature of SPI (Vicente-Serrano et al., 2010b). Thus, SPEI has been widely used in various studies for analyzing drought variability (Potop et al., 2014; Yu et al., 2014; Páscoa et al., 2017;) and severity (Liberato et al., 2017; García-Herrera et al., 2019) and several authors agree

that it is better suited to the analysis of the impacts of the increased temperature trends on drought severity than SPI (Vicente-Serrano et al., 2010a) .

A turning point on monitoring extreme events was the recent development on satellite technologies and the easier access to remote sensing data. The use of remote sensing datasets provides several advantages in the case of the analysis of phenomena with large spatial coverage, such as droughts. At the end of the 20th century and based on information from the Advanced Very

High-Resolution Radiometer (AVHRR) sensor, three satellite derived indices were proposed with the aim of monitoring the thermal and moisture conditions of vegetation: the Vegetation Condition Index (VCI)(Kogan, 1995), the Temperature Condition Index (TCI) (Kogan, 1997) and the Vegetation Health Index (VHI)(Kogan, 1997, 1998). These indices proved to be very useful in the detection and monitoring of extensive areas affected by drought, as they account for different crop sensitivities to drought, such as to moisture and thermal conditions over the vegetative cycle ( Kogan, 2001; Zarei et al., 2013;

Bokusheva et al., 2016; Ribeiro et al., 2018). Regardless of each approach's advantages and weaknesses, the use of combined drought indicators, drought indexes and satellite derived indices, which account for different time-scales of drought occurrence, have shown an added value in the performance of the crop yield simulations (Vicente-Serrano et al., 2006; Hernandez-Barrera et al., 2017; Ribeiro et al., 2018).

Even with the worldwide intense emphasis devoted to the evaluation of drought and of its impacts, the quantitative assessment

of the natural ecosystems vulnerability to drought continues to be very challenging. One of the factors which contributes to the difficulty inherent to the assessment of drought vulnerability is related with the fact that the definition of vulnerability is complex and not entirely consensual. In this context, the 5th Assessment Report of the Intergovernmental Panel on Climate Change (IPCC) (IPCC, 2014) defines vulnerability to be "the propensity or predisposition to be adversely affected. Vulnerability encompasses a variety of concepts including sensitivity or susceptibility to harm and lack of capacity to cope

and adapt".

Several approaches have been proposed to assess the agricultural vulnerability of a given region, namely based on vulnerability maps (Luers et al., 2003; O'Brien et al., 2004), composite index methods (Kim et al., 2015; Murthy et al., 2015b, 2015a; Wiréhn et al., 2015; Jiang et al., 2012i) and  Principal Component Analysis (PCA) (Li et al., 2006), among others. Conversely, Bogardi and Birkmann (2005) emphasize the need to address the components of Exposure, Sensitivity and Adaptability of



crops in vulnerability studies. Following Murthy et al. (2015b), the component of Exposure is associated with the nature, extent, duration and frequency of drought conditions over a geographic area. In contrast, Sensitivity is the degree to which the crops respond to drought conditions and it is related with cropping pattern and crop condition. Finally, the Adaptive Capacity incorporates the ability of an agricultural area to cope-up with an agricultural drough episode. This study was later reinforced by the 4th IPCC Assessment Report (IPCC, 2007) which indicated that vulnerability assessments should encompass the above

components.

Vulnerability assessment and risk management for drought events allow to mitigate the adverse effects of these events in a proactive way (Murthy et al., 2015b). Consequently, and considering 1) the significant impacts of drought events and, 2) the need for adaptation and resilience of the population against these extremes, a vulnerability assessment method is proposed here to identify the most vulnerable regions over Portugal.

The most important drought impacts reported for past events in mainland Portugal have been agricultural losses, interruptions in public water supply, and wildfires, and therefore, a way of increasing drought preparedness is to identify specific periods where water shortage is crucial for maximizing impacts (Dias et al., 2019). Therefore, in this paper we present an important application of an automatic approach which is able to identify agricultural areas which are vulnerable to drought conditions. The proposed approach is grounded on the application of a PCA to the Exposure, Sensitivity and Adaptability components to

obtain agriculture vulnerability maps to drought in mainland Portugal, using the above mentioned vulnerability components and considering that the PCA approach enables to reduce considerably the amount of input data needed for achieving drought vulnerability maps. This reduction in the number of input information is possible because the derived variables incorporate a great amount of information which is able to reduce redundancy on the input variables and summarize the most important features (Wilks, 2011). Moreover, the PCA assumes that the significant information retained by the derived variables does not

significantly change, which is a major advantage when working with observational data. Finally, to support and corroborate the PCA approach, a categorical methodology (Murthy et al., 2015a, b) was applied based on the same components used in the PCA.

This paper is organized in 5 sections. After the introduction to the proposed work, Sec. 2 briefly describes the study area and the data used to represent the components of Exposure, Sensitivity, and Adaptive Capacity. In Sec. 3 the methodology is

described. In Sec. 4 the results for the PCA and the Murthy approaches are discussed, and Sec. 5 concludes the paper.

## 2 Data

### 2.1 Study Area

The precipitation regime presents a high interannual and decadal variability over Mediterranean basin, namely in Portugal (García-Herrera et al., 2007; Trigo et al., 2013). A pronounced northwest-southeast precipitation gradient is observed over

Portugal, and annual precipitation ranges from more than 2000mm in the North to less than 500mm in the South (AEMET-IM, 2011). Moreover, most of the precipitation is concentrated in the months from October to March (García-Herrera et al.,



2007; Trigo and DaCamara, 2000). There is also a North-South temperature gradient, and the high temperatures occurring in the summer coincide with the dry season (AEMET-IM, 2011). There is a clear trend towards warmer conditions (Espírito Santo et al., 2014a), in agreement with the global warming trend, and a decrease in spring precipitation (Espírito Santo et al., 2014b).

The majority of the climate change scenarios for the Mediterranean region show a consensual evolution in the future , indicating declining precipitation and rising temperatures in southern Europe (Giorgi and Lionello, 2008; García-Ruiz et al., 2011; Mariotti et al., 2015). This combined effect of lower precipitation and higher temperatures is expected to increase the frequency, magnitude and severity of drought episodes (Vicente-Serrano et al., 2014a). These trends create a potential threat

to the agricultural sector, especially for rainfed agriculture, since its production is dependent on the precipitation regime, and thus they may affect the economic viability of some crops (Valverde et al., 2014). Previous studies focusing on drought occurrence in the Iberian Peninsula (Gouveia et al., 2009, 2017; Vicente-Serrano et al., 2014b; Páscoa et al., 2017) identified several severe/intense drought episodes, namely in 1981, 1995, 2000, 2002, 2005 and 2012. Vicente-Serrano et al. (2014b) showed that between 1961 and 2011, drought severity and the surface area affected by drought increased in the Iberian

Peninsula. On the other hand, Ribeiro et al. (2018) showed that there are anomalies in the production of cereal yields (wheat and barley) during the years 1992, 1995 and 2005, coinciding with the main drought events that affected the Iberian Peninsula (García-Herrera et al., 2007;Andrade and Belo-Pereira, 2015). Therefore, this type of study in mainland Portugal will be an added value as it will allow the identification of areas in which crops are more susceptible to drought events, allowing for better future planning.

## 2.2 Drought indicators

Drought assessment was made using two types of indicators, one calculated based on ground meteorological observations and the others based on satellite information. The first indicator was SPEI (Vicente-Serrano et al., 2010a, 2010b). SPEI was computed using monthly precipitation and reference evapotranspiration from the Climatic Research Unit (CRU) TS4.01 high-resolution gridded dataset (http://adc.nerc.ac.uk/data/cru/) with a spatial resolution of 0.5º, for the period 1901-2016 and for

temporal scales of 6 and 12 months (Harris et al., 2014). Reference evapotranspiration ($ET_0$) was estimated using the Penman-Monteith method (Monteith, 1965). SPEI was estimated using a log-logistic probability distribution, which allows for a very good fit to the series of differences between precipitation and $ET_0$ (Vicente-Serrano et al., 2010b). The parameters were estimated using the L-moment method (Russo et al., 2015).

The second type of indicator used to assess drought was VHI (Kogan, 1997), which evaluates the vegetation condition,

including the effect of humidity and temperature. VHI is calculated as the mean between VCI and TCI, and in this work was used to characterize the Sensitivity component. VHI evaluates the vegetation condition, including the effect of humidity and temperature, whilst VCI allows to identify zones of stressed vegetation related to the amount of water (moisture), and TCI allows to identify zones of vegetation in thermal stress (Kogan, 1997).





A one-month time frame could be too long to describe the vegetative cycle, as morphological changes and leaf appearances
occur every 3–7 days (Kogan, 1997). On the other hand, weather patterns change even faster, considering that an elementary
synoptic period continues for 3–5 days (Kogan, 1997). Therefore, in case of severe drought, vegetation can be desiccated in a
matter of days (Kogan, 1997). The values of VHI were produced and disseminated by NOAA
(ftp://ftp.star.nesdis.noaa.gov/pub/corp/scsb/wguo/data/VHP_4km/geo_TIFF/), with weekly frequency and 4 km of spatial
resolution. The data used in the present work covers the period 1981 to 2019.

### 2.3 Agriculture data

The    agriculture    datasets    were    extracted    from    the    National    Statistics    Institute    platform    (INE)
(https://www.ine.pt/xportal/xmain?xpid=INE&xpgid=ine_base_dados). The main agricultural crops (main grain crops, main
dry legumes, potatoes, major crops for industry, horticultural crops, main forage crops, main fresh fruits, small berry fruits,
main subtropical fruits, citrus fruits, principal nuts, vines and olive groves) production area datasets were disseminated at an
annual basis, in hectares, from 1986 to 2015, and organized in Territorial Units for Statistics (NUTs II): Norte, Centro, Área
Metropolitana de Lisboa, Alentejo e Algarve. Using the area in hectares of each NUT, this variable was converted into the
percentage of annual area of the main agricultural crops by NUT.

The percentage of irrigable area in agricultural areas was also extracted from the INE platform. Data were only available on
the years 1989, 1993, 1995, 1997, 1999, 2005, 2007, 2009 and 2013, for each NUT II.

### 2.3 Soil characterization

The Aridity Index (AI) applied in this work was proposed by the FAO (Food and Agriculture Organization of the United
Nations) (Spinoni et al., 2015). This index was calculated from the average relation between total annual precipitation (P) and
potential evapotranspiration (PET), for a period of 30 years, in this case 1971-2000. The precipitation was extracted from a
control run for the period 1971-2000 of the WRF (Weather Research and Forecasting Model) with 9 km of resolution for the
PI and forcing from the EC-EARTH (Soares et al., 2017). The PET was calculated using the Hargreaves equation (Hargreaves
and Samani, 1985) based also on the WRF precipitation and temperature data. Although the Penman-Monteith method is
generally the recommended method to estimate $ET_0$, it needs a large number of meteorological variables, and they may not all
be available. The Hargreaves method is known to provide estimates closer to the Penman-Monteith method, when compared
with other methods that require less variables (Beguería et al., 2014). Moreover, the high spatial resolution of this dataset is
advantageous, since aridity depends on variables that are sensitive to topography.

The Water Table Depth (WTD) provided by Fan et al. (2013) was used as a proxy of soil water availability over the considered
region. The authors used a groundwater model, forced with climate, terrain, sea level, and observations of WTD compiled
from government archives and literature. The WTD obtained is a mean value and it should be noted that only the time-series
longer than 4 years and with declining trends smaller than 0.6m per year were included in the model.. For Portugal, the authors
used 438 points which are not evenly distributed in the study area, but this is related with the uneven distribution of the



available stations that monitor piezometry in Portugal (Gomes Marques et al., 2018). Nonetheless, it is likely that the data is biased, and this should be taken into consideration (Fan et al., 2013).

## 2.3 Land cover

Information regarding land cover classification were obtained using the Corine Land Cover (CLC) map version 18 for the year

2006. CLC maps contain 44 land cover classes and are available with a spatial resolution of 250m. The original projection is Lambert azimuthal equal area, and so it was reprojected and resampled to match the VHI projection and spatial resolution, using a nearest neighbor interpolation.

## 3 Methodology

### 3.1 Components of vulnerability

The methodology proposed in this work relies on the use of the indicators described in the previous section. The indicators are grouped, considering the three components used to characterize vulnerability, i.e., Exposure, Sensitivity and Adaptive Capacity (Table 1).

### 3.1.1 Exposure component

The exposure of the study area to drought was characterized during the growing season of crops using SPEI at 6-month time scale, and during the hydrological year using SPEI at 12-month time scale. In the study area, the influence of drought on crops is more pronounced from January to June (Páscoa et al., 2017, Ribeiro et al., 2018), and so we used the SPEI 6 value in June. The SPEI 12 value in August was used. Although the hydrological year in Portugal ranges from October to September of the following year, we considered the period September to August (Gouveia et al., 2009), since 1981 to 2016. Maximum and

minimum values of SPEI were computed, as well as the number of months identified as extreme drought (SPEI $\leq$-2). Using SPEI 6, the number of months identified as moderate drought ($-1 \geq$ SPEI $\geq -1.49$) were also computed. The thresholds used to characterize drought intensity were proposed by Rhee and Cho (2016). The thresholds used to characterize drought intensity were proposed by Rhee and Cho (2016). The values of each variable derived from the SPEI data were spatially smoothed, using a mean filter over a user-defined rectangle, aiming to harmonize the resolution of the SPEI data with VHI data. The

indicators used for characterizing the Exposure component are shown in Figure 1.

### 3.1.2 Sensitivity component

Two metrics were applied to the VHI data during the growing season of the winter crops, i.e., period ranging between January to June (Páscoa et al., 2017, Ribeiro et al., 2018), namely the Season's Integrated (SI) and the Season's Maximum (SM) (Murthy et al., 2015b). The first metric is relative to the annual coefficient of variation (CV) of accumulated VHI values during the



growing season and the second refers to the annual CV of the maximum VHI value for the same period. In addition, in order
to characterize the frequency of droughts impacts on vegetation, the CV was computed for the number of times per year in
which VHI values were lower than 20, severe vegetation stress conditions, between during growing season; and the number
of times that VHI values were less than 40, vegetation stress conditions,  in the total data period, since September 1981 to
August 2016 (i.e., number of weeks, in the 35-year data in which VHI <40). The thresholds used were proposed by Kogan

(1998). The statistical parameter corresponding to the main agricultural crops annual area was used to characterization the type
of crop. The indicators used to simulate the Sensitivity component are shown in Figure 2.

### 3.1.3 Adaptive Capacity component

Adaptive Capacity is generally determined by the static parameters of the agro-region, i.e., parameters without intra-annual
variation (Murthy et al., 2015a). In this case, the irrigable area, the aridity index and the groundwater table depth, were

considered as static parameters. Data of irrigable area were only available for some years, by NUTs II, and so the mean was
calculated. The mean of the total agricultural area was also computed, and then used to compute the percentage of the irrigable
area in relation to the total agricultural area, making this a static parameter.

The soil does the connection between climate and crops and it can be an important factor in determining the severity of
agricultural droughts (Murthy et al., 2015b). In the present work, in order to classify the region in terms of water balance under

normal climatic conditions, the aridity index (AI) was used. Moreover, the WTD data was used to characterize the groundwater
distribution of the region, since it may be a source of water even in dry conditions. Considering that the WTD data used consists
of mean values, this is also a static parameter. Figure 3 shows the indicators used for modelling the adaptive capacity
component over the Portuguese mainland.

### 3.2 Drought vulnerability assessment

The assessment of drought vulnerability was performed using two methods: a principal component analysis, and a categorical
method. The pixels corresponding to urban and industrial zones and areas of high humidity - surface water resources - were
excluded according to CLC.  The results obtained with these methods were then compared, and the CLC classes occurring in
areas with different classification were examined. The two methods used are described in the following sections.

### 3.2.1 Principal Component Analysis

Some of the variables used to compute the different indicators of the Exposure, Sensitivity and Adaptive Capacity components
(Table 1) might correlate as they have a certain amount of overlap (Anselin and Getis, 1993; Kang et al., 2015). For this reason,
a principal components analysis was performed, since it converts potentially correlated variables into uncorrelated variables
that capture the variability in the underlying data (Abson et al., 2012). One advantage of applying PCA is its efficacy to
highlight patterns within multivariable data (Abson et al., 2012).



Each variable is standardized and therefore PCA uses orthogonal linear transformation to identify a vector in the N-dimensional space that accounts for as much of the total variability in a set of N variables as possible. The first principal component (PC) explains the higher amount of variance within the dataset (Hatcher, 1997). The second PC have two characteristics: this component will account for a maximal amount of variance in the dataset that was not accounted by the first component, and it is uncorrelated with the first component (Hatcher, 1997). Each succeeding PC accounts for as much of the remaining variability

as possible that was not accounted by the preceding components; and each one is uncorrelated with all the others PCs (Hatcher, 1997). When the original variables are correlated, then the higher order PCs will capture more of the total variability in the data than any individual original variable. Excluding the lower order PCs, the dimensionality (number of variables) of the data is reduced while minimizing the loss of information (Smith, 2002). Each PC can be related to the original variables that the PC is most influenced by through the reported principal component loading factors. Loading factors associated with each

retained PC allow the original variables to be readily associated with the resulting indices. As a result, PCA provides an approach to move from a large suite of individual indicators to a small number.

In this work, PCA was applied as a three-step process. Firstly, the PCA was applied to the indicators of the Exposure, Sensitivity and Adaptative Capacity components (Table 1), and the group of PCs that represented more than 85% of explained variance of each group of indicators was chosen to represent each Exposure, Sensitivity and Adaptative Capacity component,

as in Equation 1:

$$y_i = \sum \lambda_j * X_{i,j} , \qquad (1)$$

were $y$ is the vulnerability component, $i$ is the pixel number, $j$ is the indicator, $\lambda$ are the eigenvalues of the indicators covariance matrix and X is the principal component.

Then, a second PCA was applied to the 3 maps obtained before (Exposure, Sensitivity and Adaptative Capacity). We chose

the set of PCs that represented more than 85% of explained variance, and the final map was obtained using again Equation 1. Finally, the resultant is scaled between 0 and 1, and then divided into five drought vulnerability classes. The assignment of the obtained results to the set of five drought vulnerability classes (less and moderately vulnerable, vulnerable, highly and extremely high vulnerable) was based on the computation of the percentiles 20, 40, 60 and 80.

### 3.2.2 Categorical Method

The categorical method used in this work was based on the methodology applied by Murthy et al. (2015b) to determine the Agriculture Drought Vulnerability Index (ADVI). This crop-generic index of agriculture drought vulnerability was derived from the composite indices of Exposure, Sensitivity and Adaptative Capacity (Murthy et al., 2015b). Firstly, the differences in the units of the input indicators were normalized based on the functional relationships between indicators and respective component index (Table 1). Since exposure is associated with nature, extent, duration and frequency of drought conditions,

the contribution of SPEI values is negative to exposure, because negative values of SPEI represent drought events. In other way, the number of times that SPEI is less than -1 or -2 is the frequency of droughts and it is positively related with exposure





of agriculture crops to droughts. In the case of Sensitivity indicators (8-12 indicators, Table 1), they all have a positive relationship with the component, the higher the CV (indicators 8-10) more is the sensitivity of the agricultural area to weather variations (Murthy et al., 2015b), and the higher number of times that the VHI is less than 40 more is frequency of drought in

crops. The mean surface area used for agriculture, have a positive relationship with the sensitivity to. All of the Capacity Adaptative indicators (13-15 indicators, Table 1) are related with water in soil and as more water is available in the soil, the greater the adaptability of agricultural crops to drought (positive relationship).

An indicator, $X$, which is positively related to respective component index, is normalized using the formula:

$$X_{i-norm} = \frac{X_i - X_{min}}{X_{max} - X_{min}}, \tag{2}$$

were $X$ is the indicator, $i$ is the pixel number, $X_{min}$ is the minimum value of the indicator and $X_{max}$ is the minimum value of the indicator.

When X is negatively related with the respective component index, it is normalized by the following formula:

$$X_{i-norm} = \frac{X_{max} - X_i}{X_{max} - X_{min}} \tag{3}$$

To assess of the weights, $w$, to indicators for each component index, Murthy et al. (2015b) use the method proposed by Iyengar and Sudarshan (1982) and followed by Hiremath and Shiyani (2012):

$$w_j = \frac{c}{\sqrt{var(x_{i,j})}}, \tag{4}$$

where $c$ is a normalizing constant such that

$$c = \left[ \sum_{j=1}^{K} \frac{1}{\sqrt{var_i(x_{i,j})}} \right]^{-1} \tag{5}$$

In this case, $K$ is the number of indicators, such that $j = 1, 2, ..., K$.

For each component the composite index, $y$, is defined by the following formula:

$$y_i = \sum_{j=1}^{K} w_j x_{i,j} \tag{6}$$

Using the respective weights, the three vulnerability components were computed and scaled to range between 0–1 for easy interpretation. They were named by the authors as Exposure Index (EI), Sensitivity Index (SI) and Adaptive capacity Index

(AI). Finally, the Agricultural Drought Vulnerability Index (ADVI) was then computed as follows:

$$ADVI = EI + SI - ACI \tag{7}$$

where


$$EI = a_1 * SPEI_{12_{max}} + a_2 * SPEI_{12_{max}} + \cdots + a_7 * N^oSPEI_6 < -2 \qquad (8)$$

$$SI = b_1 * CVac_{VHI_{JanJun}} + b_2 * CVmax_{VHI_{JanJun}} + \cdots + b_5 * \overline{Surface\ Area} \qquad (9)$$

$$ACI = c_1 * \overline{Irrigable\ Area} + c_2 * AI + c_3 * WTD, \qquad (10)$$

and $a, b$ and $c$ are the weights of indicators.

After computing the ADVI map, the values of ADVI were scaled between 0 and 1, with equation 2, and the values of five classes were obtained using also the percentiles 20, 40, 60 and 80.

## 4 Results and discussion

The spatial pattern of each component of Exposure, Sensitivity and Adaptative Capacity and loading factors for each of the original variables used to build each component are showed in Figures 5 and 6, respectively. The spatial pattern associated to the Exposure component is obtained by the first 3 PCs of indicators 1-7 of Table 1, which explains 91.73% of the total variance (Figure 4). This component presents negative values on the northwest and positive values on the southeast. PC1 has negative contributions from indicators 1, 2, 4, and 5, i.e., both the maximum and the minimum values of SPEI negatively contribute to
PC1; PC2 has negative contributions from indicators 2, 5 and 7, corresponding to the minimum SPEI values and to the occurrence of severe drought; and PC3 has negative contributions from indicators 1 and 7 only. Indicators 3 and 6 only have positive contributions.

On the other hand, the Sensitivity component is represented by the first 3 PCs of the indicators 8-12 (Table 1). The ensemble of first 3 PCs of these 5 indicators explains 90.42% of the variance (Figure 4) and presents positive values in the northeastern
and southern regions, except in the extreme south. The Sensitivity component shows a spatial pattern very close to the Exposure component. In the case of this component, PC1 have positive contributions from all indicators. The indicators 8-10 have a negative contribution on PC2 and PC3. The number of times that the VHI is less than 40, indicator 11 (mean surface area used in culture, indicator 12) have a negative (positive) contribution of PC2 (PC3). The indicator 12 have a positive contribution of three PCs.

To quantify the Adaptative Capacity component the first 2 PCs were selected from the last three indicators, representing a total of 91.30% of the explained variance (Fig 4). This component shows slight negative (positive) values in Alentejo (North) region. All the indicators (mean percentage of irrigable area, aridity index and water table depth) have a positive contribution to PC1, and different contributions to PC2. Water table depth (indicator 15) shows a small positive contribution in PC1 and a high positive contribution in PC2.

A PCA on the 3 vulnerability components obtained previously was performed and the two first principal components were selected to represent vulnerability, explaining 89.12% of the total variance (Figure 7 (left panel) and Figure 8). For PC1, the Exposure and Sensitivity components present a positive contribution to the final map of vulnerability and show similar weights, whereas the Adaptative Capacity presents the opposite. PC2 shows a positive contribution of the 3 vulnerability components.





The spatial pattern in Figure 7 is characterized by maximum values in the Alentejo region and minimum values in the northwest
region, with a southwest-northeast transition zone.

Figure 7 (right panel) presents the spatial distribution of the 5 defined classes of vulnerability to drought for the main agricultural crops derived from the map obtained previously (Figure 7, left panel). Agriculture over central and north coast areas of Portugal seems to be less to moderately vulnerable to drought. The Alentejo, with the exception of the coastal region and near the Tagus river that is highly vulnerable, exhibits the higher vulnerability class (extremely high). The southwest-
northeast transition region is a vulnerable zone, as well as the northeast of Portugal.

The categorical method of Murthy et al. (2015b) was also used based on the same indicators (Table 1). Figure 9 presents the spatial pattern of Exposure, Sensitivity and Adaptative Capacity indexes obtained; the weights and the contribution (positive or negative) for each of the original 15 indicators corresponding to each index are shown in Figure 10. The map representing the Exposure of crops to drought shows higher values in the southeast and lower in the northwest regions, respectively (Figure
9, left panel).  As the negative values of SPEI represent drought events, the contribution of SPEI values is negative to exposure (Figure 10, left panel); the number of times that SPEI is less than -1 or -2 is the frequency of droughts and it is positively related with exposure of agriculture crops to droughts.  The Exposure index has a similar contribution from each indicator, being that the contribution of indicators 2 and 5 (minimum SPEI) is slightly higher (Figure 10, left panel).

Through the analysis of the spatial pattern of the Sensitivity index, it is evident that the sensitivity of crops to drought is higher
in Alentejo and northeast region and lower in the central area of Portugal (Figure 9, middle panel), showing a spatial pattern very close to the Exposure index. The SI of VHI captures most part of the total growth of the crops and the SM of VHI represents the maximum VHI of the growing season. SI and SM (indicators 8 and 9) together represent the crop vigor and its sustenance in the season (Murthy et al., 2015b). The higher the CV (indicators 8-10) more is the sensitivity of the agricultural area to weather variations (Murthy et al., 2015b). The number of times that the VHI is less than 40 represent the frequency of
drought in crops (indicator 11), and this group of indicators together with the mean surface area used for agriculture (indicator 12) have positive contribution to sensitivity of crops to drought (Figure 10, middle panel).

Additionally, the spatial distribution of the Adaptative Capacity index values presents a spatial pattern similar with the previous ones, however showing smaller values. Indicator 15 (Water Table Depth) has the highest contribution than the others. All of the indicators have a positive contribution because they are related with water in soil and as more water is available in the soil,
the greater the adaptability of agricultural crops to drought.

The spatial distribution of the ADVI was obtained through equation 7 using the obtained Exposure, Sensitivity and Adaptative Capacity indexes based on the Murthy approach. The ADVI values were scaled between 0 and 1 (Figure 11, left panel) and then converted to vulnerability classes based on the previously referred percentiles (Fig. 11, right panel).  The spatial pattern of scaled ADVI (Fig. 11, left panel) identifies areas with high and low vulnerability of agriculture to drought, namely the
central and northern coastal areas present the lowest vulnerability to drought (ADVI <Percentile 40) and Alentejo presents the highest vulnerability (ADVI> Percentile 60).



The final vulnerability maps (Figures 7 (left) and 11 (left)) are quite similar, however showing some differences (Figure 12). It should be noted that the differences found on Figure 12 correspond only to changes between two successive intermediate classes. Furthermore, both methods identify the same regions of low and extreme vulnerability, namely Minho and Alentejo, respectively. In the other vulnerability classes, few sparse differences of one class are observed, i.e., blue regions (PCA - ADVI <0) are regions in which the vulnerability class estimated by PCA is smaller than that estimated by the ADVI; and yellow regions (PCA - ADVI> 0) are areas where PCA estimates a higher vulnerability class than estimated by the ADVI. The blue regions are more common in the central and southern coastal regions of the country. In these case, the PCA method presents a lower class than the ADVI, classes of lower vulnerability predominate in this region, and the vulnerable class is dominant (Table 2). The main land cover type corresponding to these pixels is coded as heterogeneous agricultural areas, forests and schrub and/or herbaceous vegetation associations, according to Corine Land Cover 2006 (Caetano et al., 2009) (Table 3). The yellow regions are mainly in the central and northern interior of the country and correspond to higher vulnerability classes. In this case, the vulnerability class is higher as classified by PCA than by ADVI, and most of these pixels correspond to the highly vulnerable class. Regarding the types of land cover in these areas, there is a greater area of the heterogeneous agricultural areas and the shrub and/or herbaceous vegetation associations. Therefore, when comparing both approaches, we may conclude that both approaches are able to identify agricultural areas which are vulnerable to drought conditions, being the differences between them negligible.

The 24[th] report of the DROUGHT-R & SPI project (Fostering European Drought Research and Science-Policy Interfacing) (Kampragou et al., 2015) presents a study of vulnerability and risk associated with droughts (including Portugal), taking into account the components of Exposure, Sensitivity and Adaptability in some European countries. This study shows that for a vulnerability scale of 1-5 (with 1 corresponding to less vulnerable and 5 corresponding to extremely high vulnerable), Portugal presents vulnerability 4 that corresponds to a country showing high vulnerability to drought events. The present work has the advantage of having a much higher spatial resolution, allowing a regional characterization of agriculture drought vulnerability, which is crucial for regional management and a better future planning at a regional scale, namely in a climate change context. In the categorical method, the functional relationships between indicators and respective component indexes (Figure 10) is made subjectively from the *a priori* knowledge of the variable. It also should be stressed that the sign of scores and of loadings obtained from PCA is arbitrary and meaningless. It can be flipped, but only if the sign of both scores and loadings is reversed at the same time. Therefore, PCA does not necessarily define the sign of the indicators and it should be noted that the sign of the indicators in PC1 (blue bars in Figure 6) is the same of the sig defined intuitively in the Murthy method. When applying the second PCA in order to obtain the Exposure, Sensitivity and Adaptative Capacity maps, the sign of PC1 (blue bars in Figure 8) is the same of the sign of the three components on equation 7. Therefore, in order to avoid the subjective normalization of the indicators in the categorical method, the PCA could be used to normalize the indicators according to the signal of the loading factors obtained in the first PC. In some studies of vulnerability assessment, a simple and unique normalization is made for each indicator and then a PCA is applied to generate the weights. Since the first principal component contains the most information, the absolute value of the loading of the first single component is considered valid for assigning



weights (Filmer and Pritchett, 2001; Gbetibouo et al., 2010). However, the PCA approach has several advantages over the categorical method: i) the approach based on PCA does not require to perform two types of normalizations to the indicators based subjectively in the functional relationships between indicators and the respective component index; ii) the signal of the variables contributions for vulnerability components is automatically chosen by PCA.

The application of PCA to characterize and monitor drought events in Portugal is not innovative, however it has only been applied using isolated variables such as SPEI, SPI and PDSI (Martins et al., 2012; Santos et al., 2010; Vicente-Serrano, 2006). The approach used for the present work is applied to a larger and more diverse number of variables related to the components of Exposure, Sensitivity and Adaptability of the crops, which allows to evaluate the vulnerability of crops to drought. In addition, its application is supported by similar results attained by the categorical method. It also should be noted that the

proposed automatic methodology, based on PCA, could be easy extend to a broad region, such as Mediterranean basin, as does not need an a priori knowledge of regional agriculture behavior.

## 5 Conclusions

Projections of future temperature and precipitation rise in southern Europe (García-Ruiz et al., 2011) point to an increase in the vulnerability to drought (Dai, 2011; IPCC, 2007, 2014). Water resources will tend to be increasingly scarce, bringing

consequences for the production of agricultural crops (García-Ruiz et al., 2011) and, therefore, a better management of water resources will enable a better adaptation to future drought events. Agricultural drought, due to reduced moisture availability in the soil for crops, undermine food security and sustainable agriculture, giving rise to significant losses in the economy.  The present work proposes an automatic method which is able to identify the agricultural areas most vulnerable to drought, and therefore provide a tool to assist in the future planning and management in these areas. As the 4th IPCC Assessment Report

(IPCC, 2007) proposes vulnerability assessments covering the components of Exposure, Sensitivity and Adaptive Capacity of the system, this work applies a PCA to several indicators based on the definition of these components. A categorical method based in same indicators was also applied (Murthy et al., 2015b).

Both methods applied in the present study identify the Alentejo as an extremely vulnerable zone and the Minho as a less vulnerable zone. The regions that are classified by both the PCA and the categorical method as belonging to the same

vulnerability class are more likely to correspond to these two types of vulnerability. There are small differences between the methods. These differences are higher in heterogeneous agricultural areas, forests and schrub and / or herbaceous vegetation associations. However, the PCA method has some advantages over the categorical method because it can identify the sign of the indicators, not having to use the indicator-component relationship, nor calculating weights.

Alentejo is a region which should be under special attention, as this is the region where the area used for the main agricultural

crops is larger than the other regions, and it was classified by both methods as the region most vulnerable to drought. The spatial pattern of vulnerability highlights the high dependence of Portuguese agriculture on water availability (Páscoa et al., 2017; Ribeiro et al., 2018). The most vulnerable region (Alentejo) is characterized as arid and the less vulnerable regions (Central and North Coast) are characterized as humid.



Exploring the links between meteorological drought indicators and others variables that express drought impacts is considered
to be a step towards the improvement of an early warning system for seasonal drought impacts (Dias et al., 2019). The present
approach, by making use of higher resolution data, allows the identification of different zones within the country that present
different vulnerabilities. The proposed automatic technique does not need a previous knowledge about the regional relationship
between drought events and crops, as therefore the advantage of may be applied to a larger region. It is hoped that this study
will contribute to a better understanding of how and how much the agricultural sector is affected in a drought situation in order
to reduce the damages in this sector that plays an important role in the national context.

**Acknowledgments**

This work was partially supported by national funds through Fundação para a Ciência e a Tecnologia, Portugal (FCT) under
projects    IMPECAF    (PTDC/CTA-CLI/28902/2017),    IMDROFLOOD    (WaterJPI/0004/2014)    and    CLMALERT
(ERA4CS/0005/2016). The authors are in debt with Dr. P. Soares for his support with WRF data for Iberia.

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


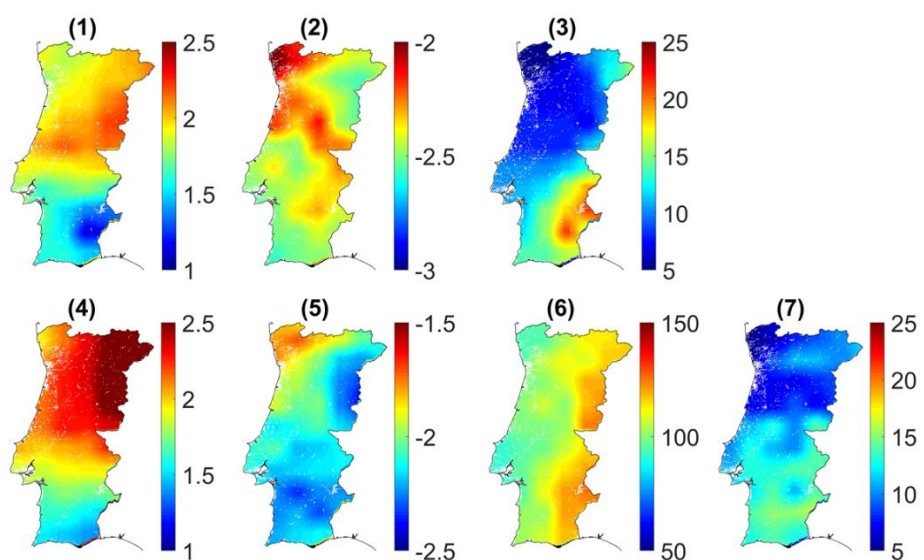

**Figure 1 – Spatial distribution of different indicators used to assess the Exposure component. The title numbers correspond to the number of the indicators in Table 1.**

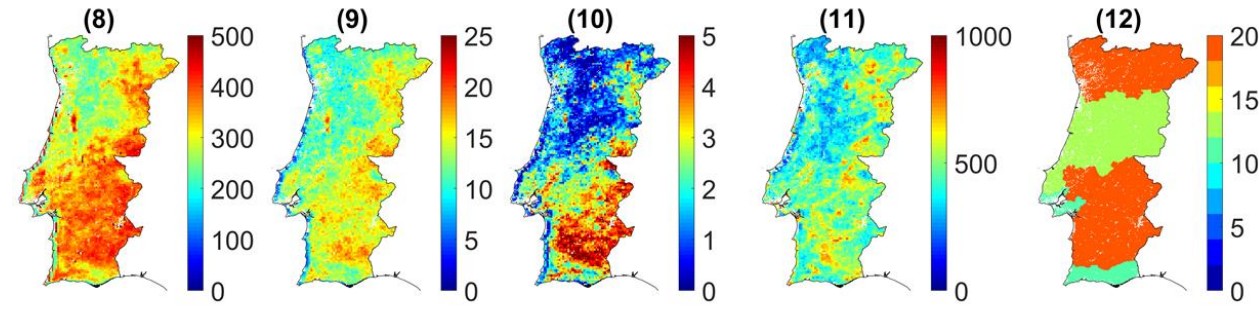


**Figure 2 – As in Figure 1 but respecting to Sensitivity component.**



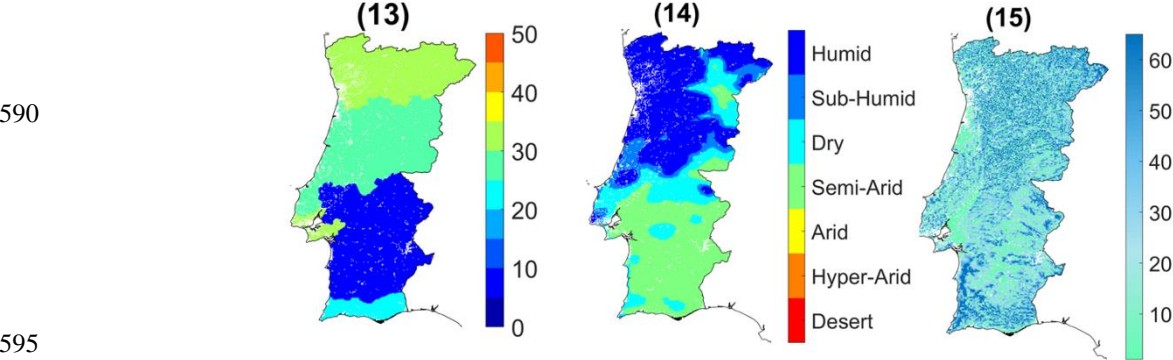

**Figure 3 – As in Figure 1, but respecting to Adaptive Capacity component.**
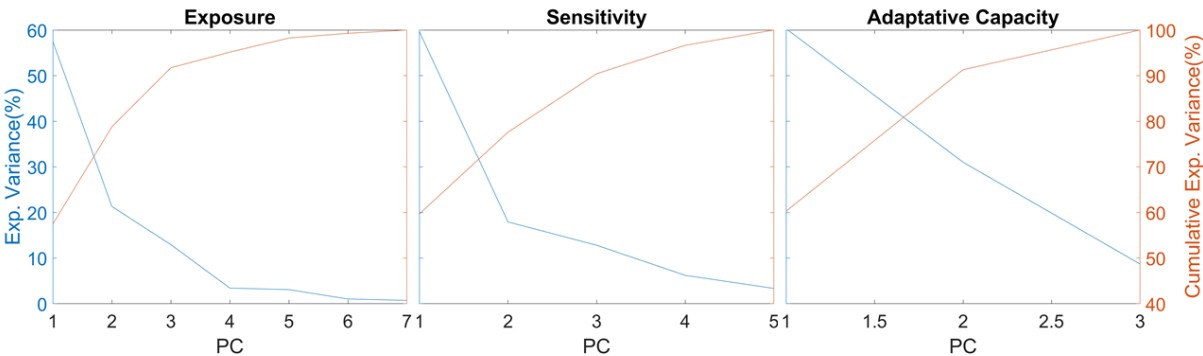

**Figure 4 - Explained variance and cumulative explained variance by each of the PCs of each component Exposure, Sensitivity and**
**Adaptative Capacity.**

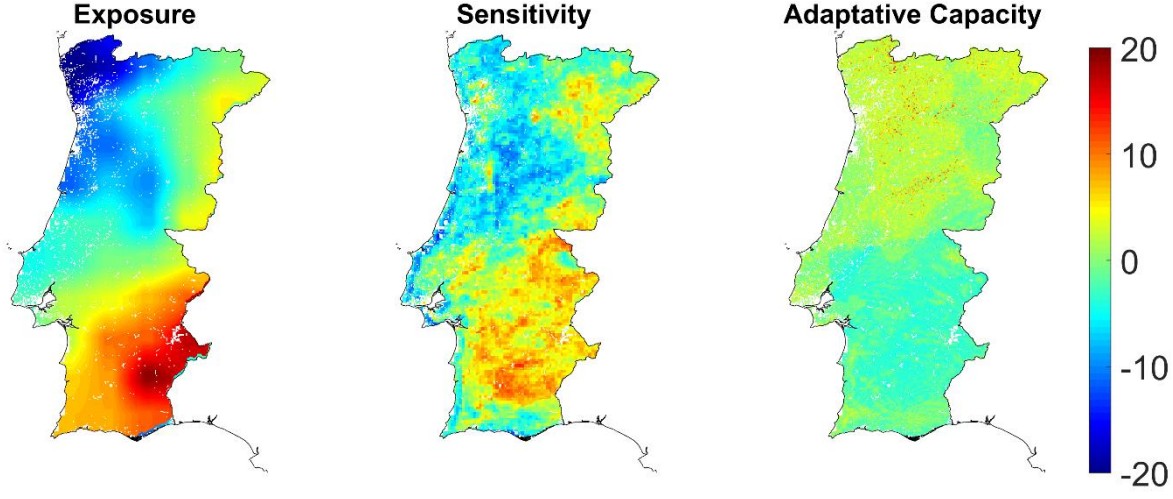

**Figure 5 – Spatial pattern for the Exposure, Sensitivity and Adaptative Capacity components, as obtained by PCA applied on the indicators presented in Table 1.**




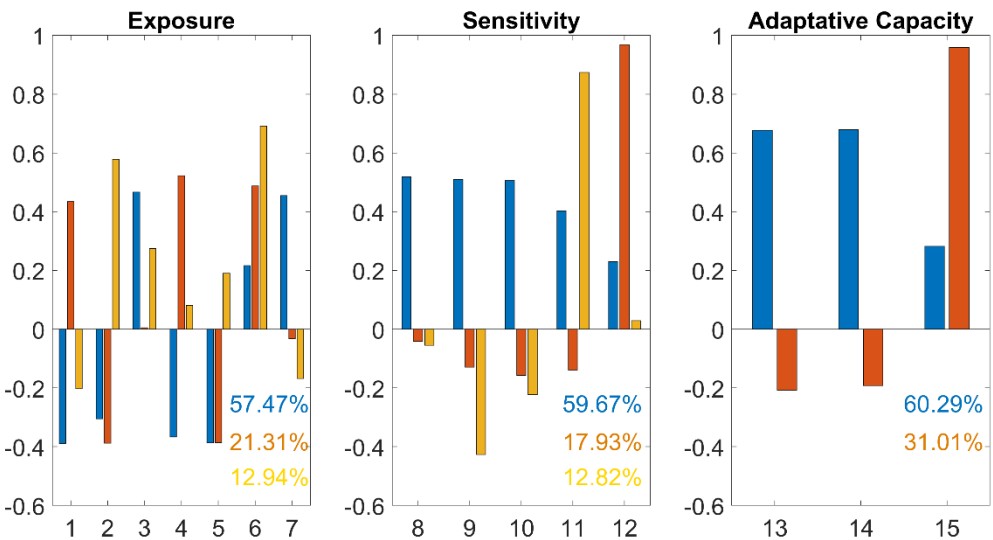

**Figure 6 - Loading factors of original 15 indicators corresponding to each PC (PC1 - blue; PC2 - orange; PC3 - yellow) used to represent the components,  X-values correspond to the original indicators described in Table 1). The percentage of variance explained for each PC is presented on down right corner.**

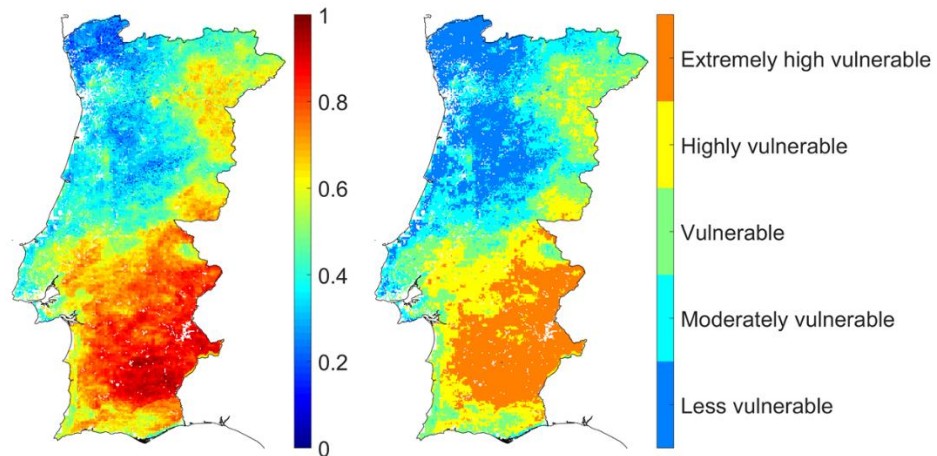

**Figure 7 - Left: Two first PC resultant from the application of the PCA to the three components presented on Figure 4 and loading factors for each component. Right: Classes of vulnerability of the main crops to drought derived from the map on the left.**


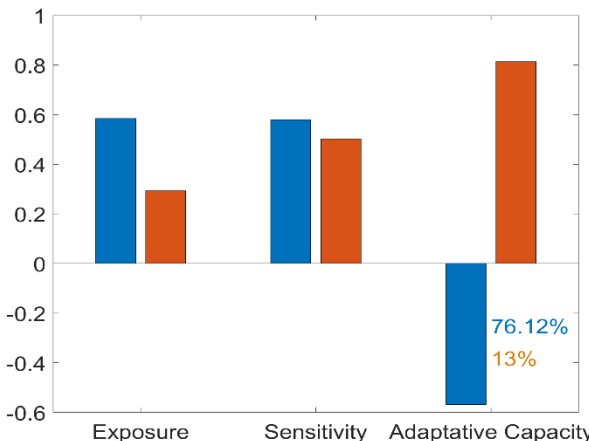

**Figure 8 - Loading factors of each PC (PC1-blue; PC2-orange) of Exposure, Sensitivity and Adaptative Capacity showed in Figure 5. The percentage of variance explained for each PC is presented on down right corner.**

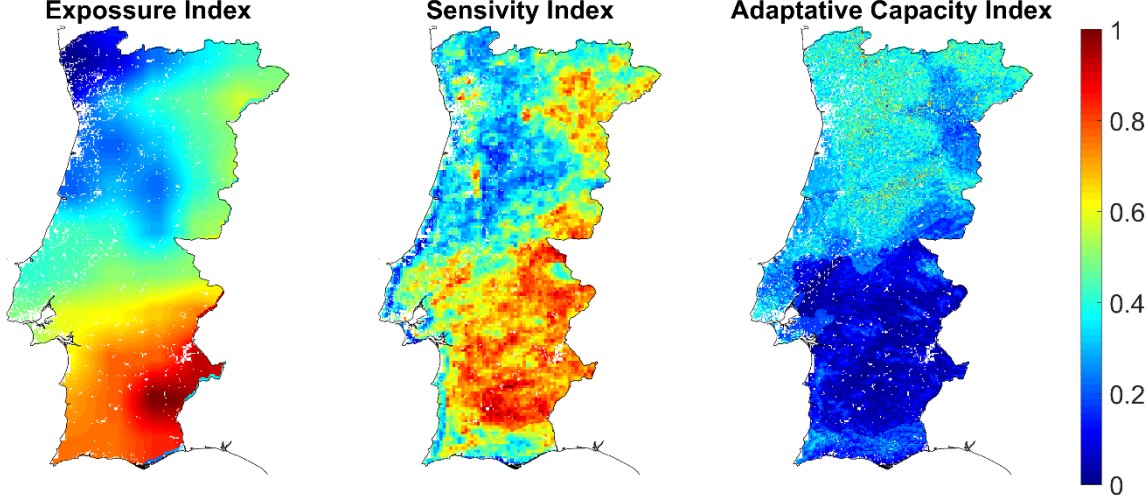

**Figure 9 - Exposure, Sensitivity and Adaptative Capacity indexes obtained with categorical method.**


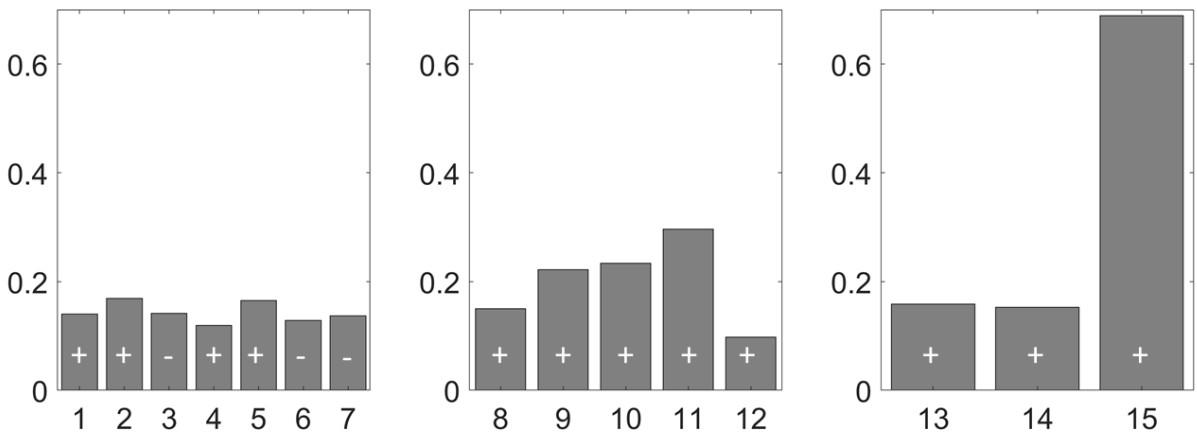

**Figure 10 – Weights associated to each of the 15 original indicators, and respective sign.**

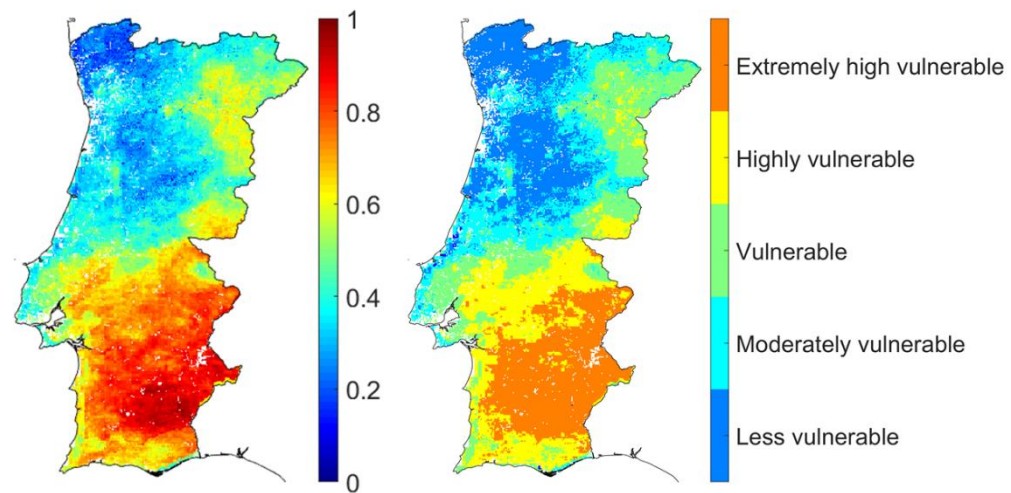

**Figure 11 - Agriculture drought vulnerability index (ADVI) obtained by the three composite indexes of Figure 8 (left). Classes of**
**vulnerability of main crops to drought derived from the ADVI map (right).**




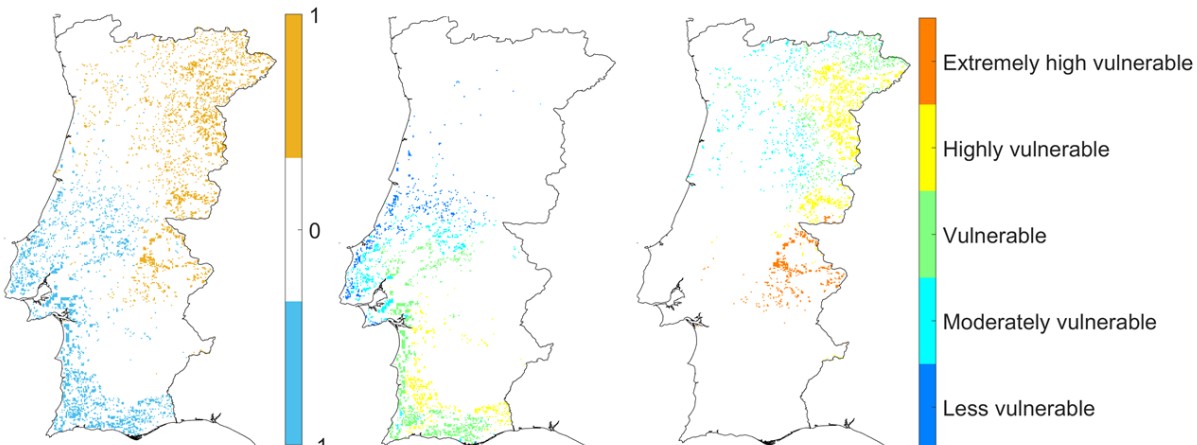

**Figure 12 – a) Differences between the vulnerability maps obtained by the PCA and categorical methods. Vulnerability classes map calculated from the PCA corresponding to the difference equal to b) -1 and c) 1.**


**Table 1 - Selected indicators**

| Component | Indicators |
|---|---|
| Exposure | 1. Maximum SPEI of 12-month scale in August |
| | 2. Minimum SPEI of 12-month scale in August |
| | 3. Number of months that the 12-month SPEI in August identifies severe drought (SPEI ≤-2) |
| | 4. Maximum SPEI of 6-month scale in June |
| | 5. Minimum SPEI of 6-month scale in June |
| | 6. Number of months that the 6-month SPEI in June identifies moderate drought (-1≥SPEI≥-1.49) |
| | 7. Number of times that the 6-month SPEI in June identifies severe drought (SPEI ≤-2) |
| Sensitivity | 8. CV of VHI accumulated between January and June |
| | 9. CV of maximum VHI between January and June |
| | 10. CV of the number of times that the VHI between January and June is less than 20 |
| | 11. Number of times that the VHI is less than 40 |
| | 12. Mean surface area used for agriculture |
| | 13. Mean percentage of irrigable area, relative to total agricultural area |





| Adaptive Capacity | 14. Aridity Index |
|---|---|
| | 15. Water Table Depth |






**Table 2 - Percentage of pixels of each vulnerability class, calculated from the PCA, corresponding to the difference equal to PCA - ADVI < 0 and PCA - ADVI >0.**

| % | PCA - ADVI < 0 | PCA – ADVI > 0 | |
|---|---|---|---|
| **Less vulnerable** | 0.92 | 0.00 | 635 |
| **Moderately vulnerable** | 1.14 | 0.92 | |
| **Vulnerable** | 2.36 | 1.14 | |
| **Highly vulnerable** | 0.98 | 2.36 | |
| **Extremely high vulnerable** | 0.00 | 0.98 | |

**Table 3 - Percentage of pixels of each corine land cover (Caetano et al., 2009), calculated from the PCA, corresponding to the difference equal to PCA - ADVI < 0 and PCA - ADVI >0.**

| Land cover class | PCA - ADVI < 0 | PCA – ADVI > 0 |
|---|---|---|
| **Arable land** | 0.47 | 0.68 |
| **Permanent crops** | 0.30 | 0.60 |
| **Pastures** | 0.05 | 0.07 |
| **Heterogeneous agricultural areas** | 1.54 | 1.67 |
| **Forests** | 1.60 | 0.61 |
| **Schrub and/or herbaceous vegetation associations** | 1.38 | 1.65 |
| **Open spaces with little or no vegetation** | 0.02 | 0.10 |