# Peer review of "Crops' exposure, sensitivity and adaptive capacity to drought occurrence"

_Natural Hazards and Earth System Sciences, 2019_

## Referee Comment (RC1) · Anonymous Referee #1 · 17 Aug 2019

The present article proposes an interesting comparison between two methodologies to identify vulnerable agricultural areas to drought in Portugal for the period 1986-2015 by using a complete dataset. The methodology relies on a PCA and the definition of three components that define the more or less vulnerability of the analysed scenario to drought, namely Exposure, Sensitivity and Adaptive Capacity. The method also includes the use of a climatic drought index (SPEI) and a satellite-derived index (VHI). In general, the paper is very well-structured, clear and helpful for management purposes, national and regional stakeholders. Not finding major issues, I highly recommend it for publishing before assessing the following minor revisions:

L41 .- Please provide more references that support this statement.

L51. – Please re-write 'indexes' as 'indices'.

[Figure]

Please clarify in section 2 the exact period of time the analysis covers. Authors mention in L-134 that the work covers the period 1981-2019 however agricultural data is only available from 1986-2015. Probably I am missing something but from my point of view it is not too clear.

L182.- Please delete the repeated sentence: ' The thresholds used to. . .'.

L184. – Are authors then using the 4km resolution as the unique one? In addition, regarding the groundwater model (L153), Is it a gridded model generated from the WTD and the other variables (climate, terrain, sea level)? In that case, Did authors use the same spatial resolution (4km)?

L261.- Please re-write 'Xmax is the minimum value of. . .' as 'Xmax is the maximum value of. . .'

L301.- Please re-write 'Fig 4' as 'Figure 4'

A suggestion for some of the figures: Please clarify the units of the scales.

---

## Referee Comment (RC2) · C. S. Murthy (Referee) · 19 Aug 2019

The current study is on drought vulnerability assessment through an index based approach and has relevance to the requirements of long term drought management. There are some important points for Authors to clarify in order to make this paper fit for publication in any international journal. Title of the paper covers only sensitivity and adaptive capacity aspects although the research work includes exposure aspect also! Any specific reason? The study has compared two methods of computing weights generation for the input indicators namely Principal Component Analysis (PCA) and Categorical method (as named by Authors). It is not categorical method – it is Variance method. It is also a statistical method and not a subjective and non-automatic method, as mentioned by the Authors. It needs to be mentioned that in both the meth-

ods weights are data driven. PCA adopts a linear approach for weights generations. Detailed information on these two methods needs to be furnished while drawing any conclusion. Why the input indicator at s.no 14 " Aridity Index" in Table 1 is shown under Adaptive Capacity (AC)? Aridity index signifies exposure to drought. Adaptive Capacity is the inherent strength of the ecosystem to cope with the drought conditions and it is generally represented by static variables.

---

## Author Comment (AC1) · 1 Oct 2019

We would like to thank Referee1 for his/her careful review and constructive feedback and suggestions. We truly believe that the changes suggested by Referee #1 will enhance the quality of the manuscript. A point-by-point response is presented below.

Referee #1: L41 .- Please provide more references that support this statement.

AR: Four new references were added as follows:

L-41: "several authors agree that it is better suited to the analysis of the impacts of the increased temperature trends on drought severity than SPI (Vicente-Serrano et al.,
2010a, 2012, 2014; Blauhut et al., 2016; Páscoa et al., 2017) ."

Two of the included references were also added to the reference list:

- Blauhut, V., Stahl, K., Stagge, J.H., Tallaksen, L.M., De Stefano, L., Vogt J.: Estimating drought risk across Europe from reported drought impacts, drought indices, and vulnerability factors, Hydrol. Earth Syst. Sci., 20 (7), 2779-2800, 2016.

- Vicente-Serrano, S.M., Begueria, S., Lorenzo-Lacruz, J., Camarero, J.J., Lopez-Moreno, J.I., ,Azorin-Molina, C., Revuelto, J., Moran-Tejeda, E., Sanchez-Lorenzo A.: Performance of drought indices for ecological, agricultural, and hydrological applications, Earth Interact., 16 , 10, 10.1175/2012ei000434.1, 2012.

Referee #1: L51. – Please re-write 'indexes' as 'indices'.

AR: Changed accordingly.  L-53: "drought indicators, drought indices and satellite-derived indices, which account for different time-scales of drought occurrence"

Referee #1: Please clarify in section 2 the exact period of time the analysis covers. Authors mention in L-134 that the work covers the period 1981-2019 however agricultural data is only available from 1986-2015. Probably I am missing something but from my point of view it is not too clear.

AR: Available SPEI data cover the period 1901-2016 (L-121), while VHI data covers the period 1981-2019 (L-136). In the case of these two variables, the period analyzed in the present study was 1981-2016 (L-181, L-196). For the remaining variables, only

the available data period was used.

Referee #1: L182.- Please delete the repeated sentence: ' The thresholds used to: : :'.

AR: It was deleted.

Referee #1: L184. – Are authors then using the 4km resolution as the unique one? In addition, regarding the groundwater model (L153), Is it a gridded model generated from the WTD and the other variables (climate, terrain, sea level)? In that case, Did authors use the same spatial resolution (4km)?

AR: The reviewer is correct. Different resolutions were used for each variable:

SPEI: L-121, "spatial resolution of 0.5o"

VHI: L-135, "4 km of spatial resolution"

Agriculture datasets: L-152, "Territorial Units for Statistics (NUTs II)"

Aridity Index (AI): L-140, "9 km of resolution"

Water Table Depth (WTD): L-159, 30arc-second

Afterwards, and in order to allow data manipulation, all the datasets were resampled without interpolation to the VHI resolution.

Referee #1: L261: Please re-write 'Xmax is the minimum value of: : :' as 'Xmax is the maximum value of: : :'

AR: It was corrected.

L-264: "were X is the indicator, i is the pixel number, $X_{min}$ is the minimum value of the indicator and $X_{max}$ is the maximum value of the indicator."

Referee #1: L301.- Please re-write 'Fig 4' as 'Figure 4'

AR: Changed accordingly.

L-304: "of the explained variance (Figure 4)."

Referee #1: A suggestion for some of the figures: Please clarify the units of the scales.

AR: In order to accommodate the reviewer comment but without adding more information to the figures a new column was added to table 1 which includes information on the units of each variable used.

---

## Author Comment (AC2) · 1 Oct 2019

We would like to thank C. S. Murthy for his careful review and constructive feedback, and also for the opportunity to engage in a stimulating discussion. We truly believe that this process will enhance and clarify the paper's content. A point-by-point response will follow.

C. S. Murthy: Title of the paper covers only sensitivity and adaptive capacity aspects although the research work includes exposure aspect also! Any specific reason?

AR: In both methods used in this study, Exposure, Sensitivity and Adaptability have the

same contribution to the final manuscript. Therefore, the title was changed to include the Exposure component:

"Crops' exposure, sensitivity and adaptive capacity to drought occurrence"

C. S. Murthy: The study has compared two methods of computing weights generation for the input indicators namely Principal Component Analysis (PCA) and Categorical method (as named by Authors). It is not categorical method – it is Variance method. It is also a statistical method and not a subjective and non-automatic method, as mentioned by the Authors. It needs to be mentioned that in both the methods weights are data driven. PCA adopts a linear approach for weights generations. Detailed information on these two methods needs to be furnished while drawing any conclusion.

AR: We understand the reviewer concern. However, we would like to stress that the subjectivity of the method does not rely on the computation of the weights, but it is related with the functional relationships between the indicators and the respective component index, i.e., the sign (positive or negative) of the contribution of each variable must be given according to the a priori knowledge of the variable. In any case, we fully agree with the reviewer and the reference to the Murthy et al. (2015a, b) method as "categorical method" was changed to "variance method" throughout the text.

L-250: "Firstly, the differences in the units of the input indicators were normalized based on the functional relationships between indicators and respective component index (Table 1)."

In the case of the PCA method it is not necessary to choose this sign or to calculate the weights.

C. S. Murthy: Why the input indicator at s.no 14 "Aridity Index" in Table 1 is shown under Adaptive Capacity (AC)? Aridity index signifies exposure to drought. Adaptive Capacity is the inherent strength of the ecosystem to cope with the drought conditions and it is generally represented by static variables.

A.R: The reviewer is completely right, and we would like to thank him for the comment. Therefore, all calculations have been redone to consider this change. In the present form, eight variables were considered in Exposure Component and only two in Adaptative Capacity Component (new Table 1). As a result, all the figures were redone. Particularly, figures 6 and 10 were changed for a better understanding by the reader.